# Microglia have limited influence on early prion pathogenesis, clearance, or replication

Brent Race[1]*, Katie Williams[1], Chase Baune[1], James F. Striebel[1], Dan Long[2], Tina Thomas[2], Lori Lubke[1], Bruce Chesebro[1], James A. Carroll[1]*

1 Laboratory of Persistent Viral Diseases, Rocky Mountain Laboratories, National Institute of Allergy and Infectious Diseases, National Institutes of Health, Hamilton, Montana, United States of America, 2 Rocky Mountain Veterinary Branch, Rocky Mountain Laboratories, National Institute of Allergy and Infectious Diseases, National Institutes of Health, Hamilton, Montana, United States of America

* raceb@niaid.nih.gov (BR); carrollja2@niaid.nih.gov (JAC)

**Data Availability Statement:** All relevant data are within the paper and its Supporting information files.

## Abstract

Microglia (MG) are critical to host defense during prion infection, but the mechanism(s) of this neuroprotection are poorly understood. To better examine the influence of MG during prion infection, we reduced MG in the brains of C57BL/10 mice using PLX5622 and assessed prion clearance and replication using multiple approaches that included bioassay, immunohistochemistry, and Real-Time Quaking Inducted Conversion (RT-QuIC). We also utilized a strategy of intermittent PLX5622 treatments to reduce MG and allow MG repopulation to test whether new MG could alter prion disease progress. Lastly, we investigated the influence of MG using tga20 mice, a rapid prion model that accumulates fewer pathological features and less PrPres in the infected brain. In C57BL/10 mice we found that MG were excluded from the inoculation site early after infection, but Iba1 positive infiltrating monocytes/macrophage were present. Reducing MG in the brain prior to prion inoculation did not increase susceptibility to prion infection. Short intermittent treatments with PLX5622 in prion infected C57BL/10 mice after 80 dpi were unsuccessful at altering the MG population, gliosis, or survival. Additionally, MG depletion using PLX5622 in tga20 mice had only a minor impact on prion pathogenesis, indicating that the presence of MG might be less important in this fast model with less prion accumulation. In contrast to the benefits of MG against prion disease in late stages of disease, our current experiments suggest MG do not play a role in early prion pathogenesis, clearance, or replication.

## Introduction

Prion diseases are a group of transmissible, slow progressing, invariably fatal brain diseases of humans and animals that are caused by the misfolding of a normal host cellular protein (PrP$^C$) into an infectious conformer (PrPSc) [1]. Misfolded PrPSc leads to recruitment and conversion of PrP$^C$ to more PrPSc through a process called seeded polymerization and to the intercellular spread of PrPSc. Within the central nervous system (CNS), this autocatalytic seeded polymerization can cause the deposition of abnormally folded protease-resistant PrP (PrPres),

**Funding:** This research was supported by the Intramural Research Program of the NIH, National Institute of Allergy and Infectious Diseases. The funders had no role in study design, data collection and analysis, decision to publish, or preparation of the manuscript. Compound PLX5622 was kindly provided by Plexxikon Inc.

**Competing interests:** The authors declare that no competing interest exist.

glial activation, vacuoles in the gray matter, neuroinflammation, and neurodegeneration [2–6]. Several of the pathological features observed during prion disease (i.e. gliosis, neuroinflammation, and neurodegeneration) are shared with other misfolded protein disorders including Alzheimer's, Parkinson's, and Huntington's diseases [7–12]. Therefore, the study of prionopathies could result in insights into these and other neurodegenerative disorders.

Microglia (MG) are self-renewing glia cells that are derived early during embryogenesis from erythro-myeloid progenitors [13–15]. These specialized glia are involved in diverse functions that include defense against infection, secretion of cytokines, neurodevelopment, phagocytosis of dead/dying cells, and maintenance of homeostasis in the brain (reviewed in [16–18]). In some neurodegenerative disease models, activated microglia can adopt a disease-associated transcriptional phenotype, abbreviated MGnD (microglia neurodegenerative) [19] or DAM (disease-associated microglia) [20], and studies have shown that chronic microglial activation can lead to neuropathology. We have shown that the expression pattern of MG-associated genes in the prion-infected brain has similarities and some differences from that found in other models of neurodegenerative disease [21], where despite long-term wide-spread microglia activation in prion disease there is little evidence to support the full transcriptional phenotype identified in MGnD or DAM subpopulations.

To better understand the impact of microglia on neuroinflammation and neurodegeneration during disease, several studies have been performed using colony-stimulating factor 1 receptor (CSF-1R) inhibitors to reduce the number of microglia in the CNS [21–25]. Microglia are reliant on continual signaling through CSF-1R, a tyrosine kinase receptor that is stimulated by CSF-1 and interleukin-34, for survival [24, 26]. Oral administration of CSF-1R inhibitors PLX5622 and PLX3397 to mice can reduce the microglia in the CNS by 90% in as little as 7 days through caspase-3 activation and apoptosis [23–25]. While other groups have demonstrated that microglia ablation in animal models of Alzheimer's disease reduces the signs of disease [27–29], we have demonstrated that reducing microglia in the CNS of mice using the CSF-1R inhibitor PLX5622 accelerates prion disease [22]. Thus, activated microglia are critical to host defense during prion infection, and their reduction results in greater PrPSc accumulation, early vacuolation, and intensified astrogliosis [21, 22, 30].

Because microglia are functionally important to host protection and their presence significantly extends survival time upwards of 30 days after prion infection [22], we began a series of studies to further dissect the influence of microglia during prion infection in mice. First, we assessed the involvement of microglia in early prion clearance after inoculation directly into the brain. Secondly, we investigated the ability of microglia repopulation following repeated short intervals of treatment with PLX5622 to positively affect prion pathology. Lastly, we studied the competence of microglia to influence prion disease in a rapid model of disease using tga20 mice that overexpress *Prnp* but ultimately have a shorter duration to clinical disease, less gliosis, and lower amounts of PrPSc accumulation relative to similarly infected wild-type mice [31–33].

## Results

### MG role in early prion disease pathogenesis

Our previous studies treating C57BL/10 mice with CSF-1R inhibitor PLX5622 beginning at either 14 or 80 dpi and continuing through terminal disease indicated that MG played a major role in prion disease survival following intracerebral inoculation of three distinct scrapie strains: RML, 22L and ME7 [22]. However, it remained unclear whether MG influenced the earliest stages of prion disease pathogenesis. We designed two experiments to explore the role of MG in the early events of scrapie infection. The first experiment directly measured

differences in susceptibility to prion infection in mice with MG or without MG. The second experiment was designed to visualize early cellular events of prion infection using immunohistochemistry and to quantitatively measure prion seeding activity using the RT-QuIC assay. We selected 22L scrapie for these experiments, as 22L is our highest titer mouse adapted scrapie, and previous studies have shown it is much easier to detect near the needle track at early time-points following inoculation [34].

To test the role of MG at the initial exposure to prion disease, we performed a limiting dilution, end-point titration to compare prion susceptibility in MG deficient C57BL/10 mice compared to normal C57BL/10 mice at the time of inoculation. MG were depleted from mice by initiating PLX5622 treatment 25 days prior to inoculation with prions (Fig 1A). Following 22L scrapie inoculation, PLX5622 was continued for an additional 60 days in the treated mice to maintain MG deficiency during the early stages of disease. Mice were euthanized when they developed end-stage signs of prion disease, or at the termination of the experiment 400 days post-inoculation. MG depleted mice were slightly more susceptible to the 22L scrapie stock compared to undepleted controls, with calculated $LD_{50}$ infectivity levels per gram of scrapie brain stock of $10^{9.77\pm0.22}$ and $10^{9.35\pm0.21}$ respectively (Fig 1B). Thus, a reduction in MG at the time of prion exposure did not have a major impact on susceptibility to disease.

To explore the role of MG during acute exposure to scrapie infection we performed stereotactic prion inoculation into the striatum. Seven days prior to inoculation and throughout the course of the experiment (days -7 up to +7), mice that were designated for MG depletion were treated with PLX5622. On day 0, each side of the brain was inoculated with 0.5 μL volumes of 10% 22L scrapie brain stock or normal brain homogenate (NBH). At 3 and 7 dpi, groups of mice were euthanized. One half of the brain was collected for histopathology analysis and a

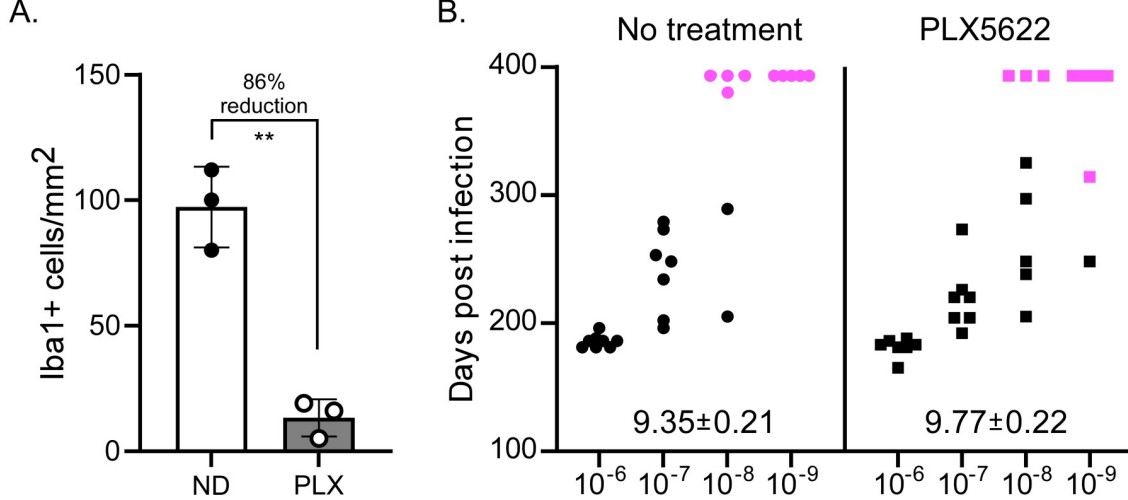

**Fig 1. End-point titration of 22L scrapie in untreated and PLX5622 treated mice.** PLX5622 was provided continuously from 25 days prior to infection until 60 days after 22L infection. At day 0, serial ten-fold dilutions of 22L scrapie were injected into untreated or PLX5622 treated (MG reduced) mice (N = 5–8 mice /group). Mice were euthanized when they developed clinical signs of scrapie, or at 400 days post-infection. Brains from mice were screened for PrPres by immunoblot to confirm infection. In panel A we assessed the effectiveness of PLX5622 treatment on selected mice from this experimental cohort. Sections of cortex were stained for Iba1, and positive cells were counted manually. Solid circles indicate the number of Iba1+ cells in individual mice with no drug (ND) and the white column is the mean. Open circles indicate the number of Iba1+ cells in individual mice treated with PLX5622 (PLX) and the grey column indicates the mean. The bars in each column represent the standard deviation. In panel B we show the end-point titration of 22L scrapie in untreated and PLX5622 treated mice. ** indicates a P value ≤ 0.01. Black symbols indicate scrapie positive mice, pink symbols indicate scrapie negative. Each symbol represents one mouse. The log LD50 for each group is shown just above the x-axis for each group and was determined using the Spearman-Karber formula.

circular biopsy around the needle track was collected from the second half of the brain to be used for prion quantification by RT-QuIC assay.

To analyze histopathology, we performed serial sections through the needle track and performed IHC specific for either microglia, macrophages and brain infiltrating monocytes (Iba1), microglia only (TMEM119), astrocytes (anti-GFAP) or prion protein (D13). At 3 dpi, all mice inoculated with either 22L or NBH had abundant Iba1 positive cells (red) associated with the needle track damage (Fig 2A–2D). This response appeared to be due to the needle trauma rather than a response to scrapie prions, since no differences in the needle track region were seen between 22L and NBH infected mice (Fig 2A and 2C). Interestingly, even the mice with MG depletion appeared to have equally robust Iba1 staining associated with the needle track (Fig 2B and 2D). We suspected these Iba1 positive cells may not be MG but could instead be Iba1 expressing macrophages or monocytes recruited into the brain post-inoculation. To test this, we used a second antibody, (TMEM119) to specifically indicate MG. IHC with TMEM119 (brown) confirmed that the cells around the needle track were not MG, but likely macrophages since they only stained with Iba1. Closer inspection of the needle track region at higher magnification showed that many of the suspected macrophages contained phagocytosed red blood cells (Fig 2I). Surprisingly, MG associated with the needle tracks were infrequent, even in the non-depleted mice (Fig 2E & 2G). In the remaining brain tissue, TMEM119 staining was similar to Iba1 staining, and confirmed that MG were reduced in the PLX5622 treated mice (compare frequency of brown cells in Fig 2E, 2G to 2F and 2H).

In addition to the histopathology, we also used the RT-QuIC assay to quantify prion seeding activity present in the area immediately around the needle track at early time-points in both MG depleted and undepleted mice. We first confirmed that MG were reduced in PLX5622 treated mice after 7, 77, and 112 days of treatment (Fig 3A and 3B, left graphs). Needle track samples from each mouse at 3 and 7 dpi were tested by RT-QuIC and a prion seeding dose 50 ($SD_{50}$) was calculated (Fig 3A, right graph). We found no differences in the amount of prion seeding present in the brains from PLX5622 treated, MG depleted mice compared to undepleted controls at either 3 or 7 dpi (Fig 3A). This result was not unexpected, since histology on these mice indicated that MG do not migrate into the needle track (Fig 2) at the prion inoculation site. Thus, MG are not in high numbers after 3 dpi at the site of inoculation in the striatum, and their reduction in the brain does not affect the prion $SD_{50}$ at 3 or 7 dpi. However, we were surprised to see that at later time-points of 77 and 112 dpi, there was still no difference in prion $SD_{50s}$ between PLX5622 treated mice and untreated mice (Fig 3B). These data suggested that any protective effect provided by removal of prions must be occurring in only the latest stages of disease.

## Intermittent PLX5622 treatments during the clinical phase of scrapie infection

Following termination of PLX5622 treatment, new MG return rapidly, and these repopulating MG have been shown to be phenotypically similar to MG present in young mice and are proficient at reversing several age-related synaptic and neuronal deficits [35–38]. If MG were becoming dysfunctional during the course of disease, we hypothesized that eliminating existing MG and allowing for repopulation might optimize the quality of MG in the brain. Rejuvenated healthy MG would then be available to remove PrPSc, and potentially delay onset of prion disease. Three PLX5622 treatment regimens were tested; a single, one week treatment (pulse) of PLX5622 beginning at 80dpi, two separate weeks of treatment (beginning at 80dpi and 101dpi) followed by two weeks of unmedicated feed, or three separate weeks of treatment (beginning at 80dpi, 101dpi, and 122dpi) each followed by two weeks of unmedicated feed (Fig

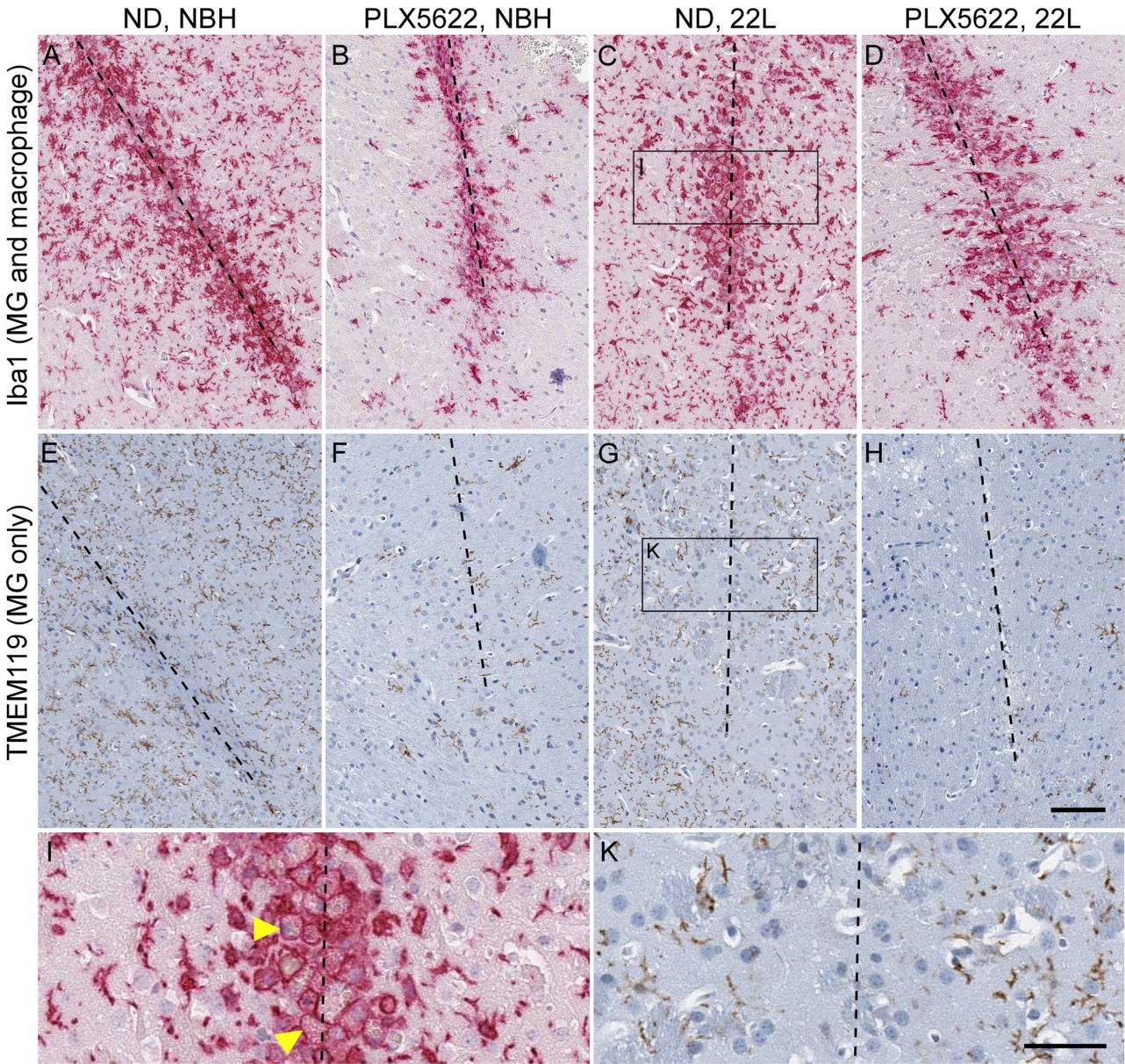

**Fig 2. Immunohistopathology associated with the needle track three days post-inoculation.** Mice were treated with PLX5622 from -7 (PLX5622) or untreated (ND), inoculated stereotactically with either normal brain homogenate (NBH) or 22L mouse scrapie in the striatum on day 0, and euthanized for analysis on day 3. The dashed line in each panel indicates the approximate location of the needle track. The antibodies used for immunohistochemistry are indicated on the far left. Anti-Iba1 is present on both microglia and macrophages while TMEM119 stains only microglia. By comparing the location of the Iba1 positive cells in the top row to the TMEM119 stained cells (microglia) in the second row, it appears nearly all the cells responding to the needle track trauma are macrophages. Also note the reduced number of TMEM119 positive cells in the PLX5622 treated mice (compare panels E&G to F&H). The rectangular insets shown in panels C and G are enlarged in panels I and K respectively. The yellow arrowheads in panel I point to examples of macrophages full of red blood cells. The scale bar in panel H is 100 μm and applies to panels A-H. The scale bar in panel K is 50 μm and applies to panels I and K.

4A). At the conclusion of each treatment week, a subset of mice was euthanized to measure MG populations following PLX5622 treatment. Following the final treatment, all the mice were monitored for clinical signs of prion disease, and mice were euthanized when they reached near-terminal disease. We found no difference in survival between the untreated mice

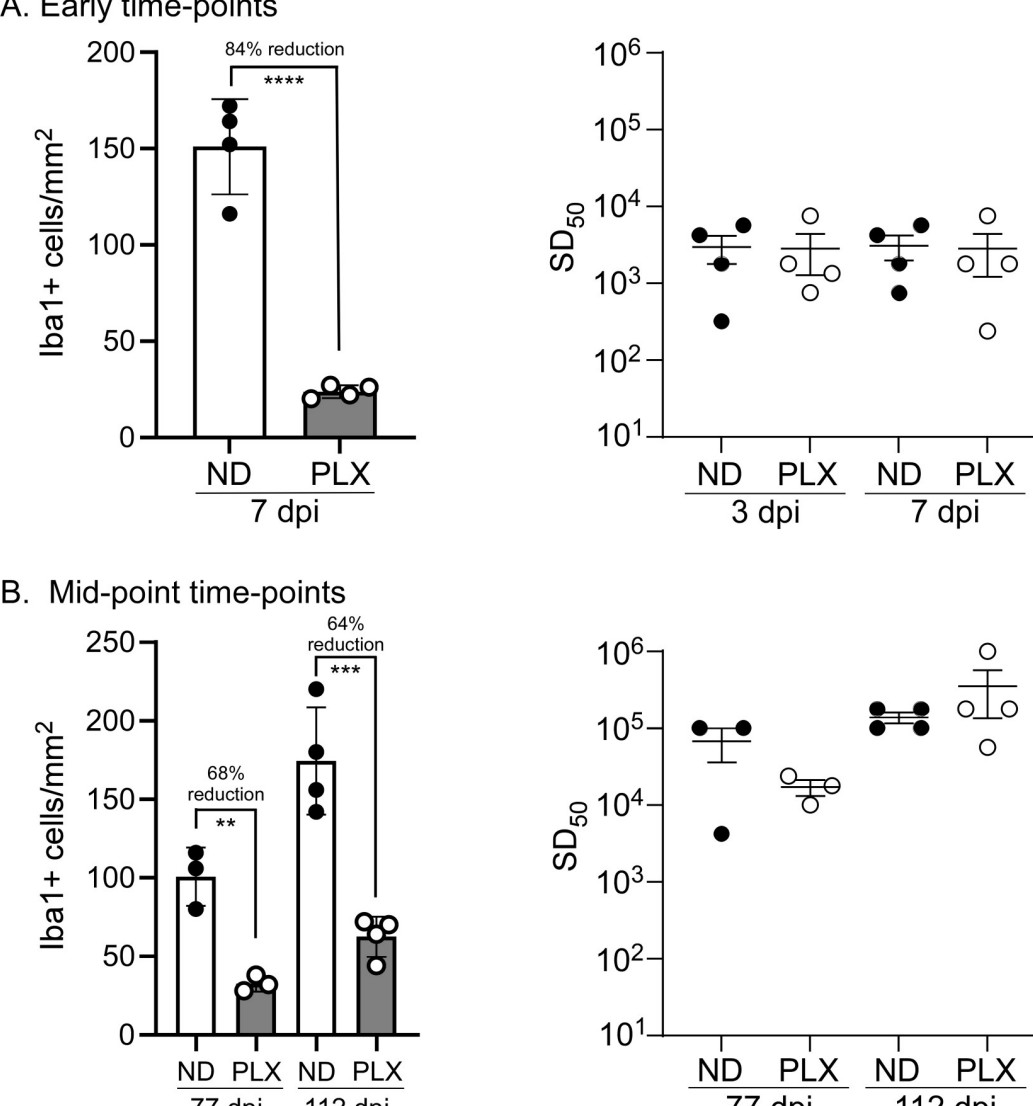

**Fig 3. Microglial depletion and RT-QuIC seeding dose determination at mid-points of infection.** Mice were treated with PLX5622 at day -7 and discontinued at day 3 or 7 (panel A) or discontinued at day 77 or 112 (panel B) when mice were euthanized for analysis. Sections of cortex were stained for Iba1, and positive cells were counted manually (panels A and B, left graphs). Solid circles indicate the number of Iba1+ cells in individual mice with no drug (ND) and the white column is the mean. Open circles indicate the number of Iba1+ cells in individual mice treated with PLX5622 (PLX) and the grey column indicates the mean. The bars in each column represent the standard deviation. ** indicates a P value $\leq$ 0.01, *** indicates a P value $\leq$ 0.001, **** indicates a P value $\leq$ 0.0001. The RT-QuIC seeding dose 50 ($SD_{50}$) was assessed at early and mid-timepoints following 22L inoculation of untreated (ND, solid circles) and PLX5622 (PLX, open circles) treated mice (panels A and B, right graphs). Serial half-log dilutions were tested by RT-QuIC to determine a single $SD_{50}$ value for each mouse. No significant differences were measured between ND and PLX mouse groups at any individual time-point (unpaired T-test).

and any of the three PLX5622 medicated groups (Fig 4B). We were initially uncertain as to why there was no difference among the treatment groups. To confirm that MG populations had in fact been reduced following PLX5622 treatments, we performed IHC staining for microglia on brain tissue from mice collected immediately post PLX5622 dosing. Unfortunately, the MG in brains from prion infected, PLX5622 treated groups were only slightly

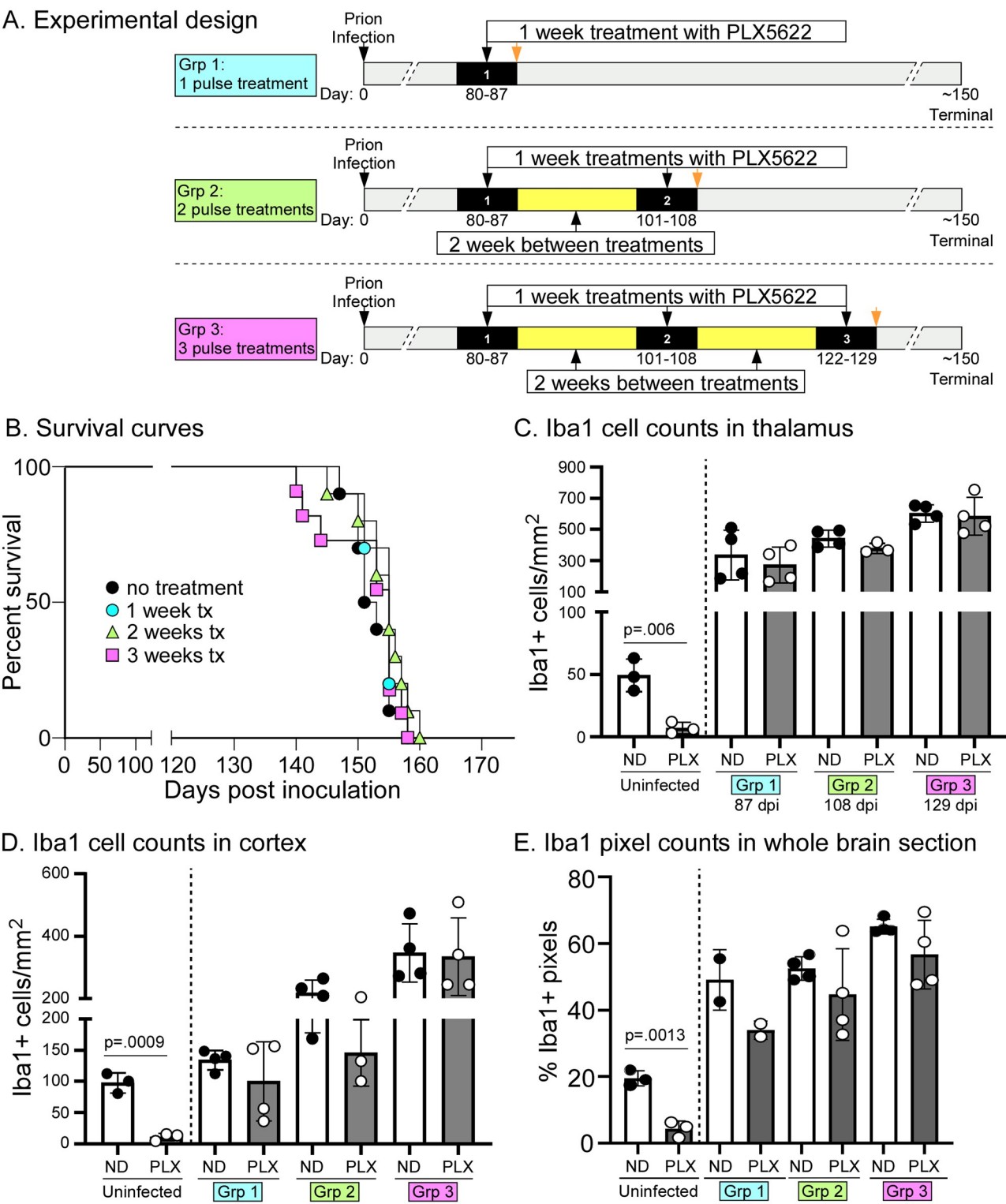

**Fig 4. Intermittent PLX5622 administration during scrapie infection. A**. Cartoon showing the different treatment groups. **B**. Survival curves following infection with 22L scrapie. Mice were untreated or treated with PLX5622 for either one week, two weeks or three weeks as described in panel A. No differences were seen between any of the groups (Mantel-Cox, Log-rank test). Each survival group contained 9–11 mice. **C, D, E**. Iba1 positive cell or pixel count comparisons in thalamus (C), cortex (D) or whole brain (E) between untreated (ND) and PLX5622 treated (PLX) mice at 87, 108 and 129 dpi. These time points were selected to correspond with cessation of PLX5622 treatment in the three different PLX treatment groups. Untreated

and PLX5622 treated mice were compared using an unpaired T-test for each experimental pair. Only the uninfected, PLX5622 treated mice were significantly different than untreated, uninfected mice. The uninfected PLX5622 treated mice were included only as a drug efficacy control and were treated with PLX5622 for 19 days concurrent with the time period of the scrapie infected PLX5622 treated mice, using the same lot of medicated feed.

reduced compared to the untreated, prion infected matched controls (Fig 4C–4E). A slight reduction in MG was notable in the thalamus, especially as disease progressed (Fig 4C), but was not statistically significant. Analysis of the cortex and sagittal sections of whole brain also showed a widespread lack of MG reduction with intermittent 7-day PLX5622 treatments (Fig 4D and 4E). Thus, short-term treatment with PLX5622 during the later stages of prion disease was ineffective at significantly reducing MG or altering survival times.

## The role of MG in a prion disease model with lower PrPres accumulation

In our previous study, we found that long-term treatment with PLX5622 reduced incubation periods of RML scrapie in C57BL/10 mice by 31–33 days [22]. PLX5622 treated mice also accumulated PrPres much faster. One hypothesis was that reducing MG might lead to dysregulation of the clearance mechanisms available to eradicate accumulating PrPSc in the brain. To explore this idea further, we tested the effect of PLX5622 treatment in RML-scrapie infected tga20 mice. Relative to C57BL/10 mice, tga20 mice overexpress PrPC by 6 to 10-fold and succumb to prion disease more quickly [31, 32]. However, at the end-stages of clinical RML-scrapie tga20 mouse brains accumulated only about 8% the amount of PrPres compared to RML-infected C57BL/10 mice (Fig 5A). We hypothesized that if the protective effect observed in mice with normal MG levels was primarily due to PrPres removal, a mouse model with minimal PrPres accumulation may not show a strong affect following PLX5622 treatment.

Tga20 mice were infected with either 1% or 0.001% RML brain homogenate stock and PLX5622 treatment was initiated at 14 days post inoculation and continued for the duration of the experiment. Mice were euthanized when they reached a near terminal stage of disease. To confirm that MG populations were reduced in tga20 mice treated with PLX5622, we performed IHC staining for microglia. Pixel counts for Iba1 positive cells were performed on sagittal sections of whole brain tissue from uninfected and prion-infected mice that were either PLX5622 treated or untreated. PLX5622 treatment was successful in reducing the number of MG in both scrapie-infected and uninfected tga20 mice (Fig 5B). Interestingly, MG in tga20 mice did not show evidence of breaking through PLX5622 suppression during late prion disease, as we previously observed in our C57BL/10 model with long-term PLX5622 treatment [22].

Similar survival results were found in both inoculation experiments, where mice treated with PLX5622 succumbed to prion disease slightly faster than untreated mice (Fig 5C and 5D). The decrease in survival times were subtle in each experiment, with only the groups infected with 1% RML being deemed significant by log-rank analysis (P = 0.0275). The average reduction of survival in both experiments (3.5 and 5.25 days) was much less than the previously observed 31-day reduction in PLX5622 treated C57BL/10 mice [22]. The 31-day reduction was approximately 20% of the untreated C57BL/10 incubation periods compared to an only 5–6% reduction in the current tga20 experiments. Thus, the presence of MG offered only a slight benefit in survival times when a rapid prion disease mouse model was employed, supporting our hypothesis that a mouse model with minimal PrPres accumulation may not show a strong affect following PLX5622 treatment.

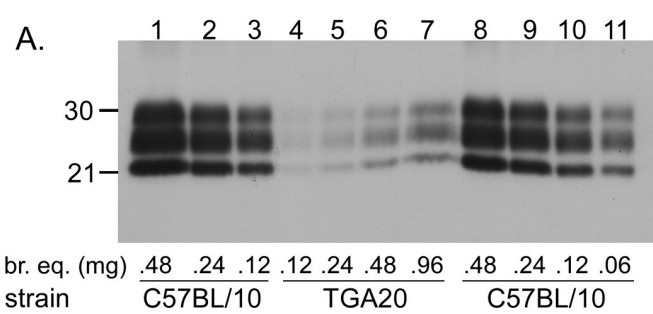

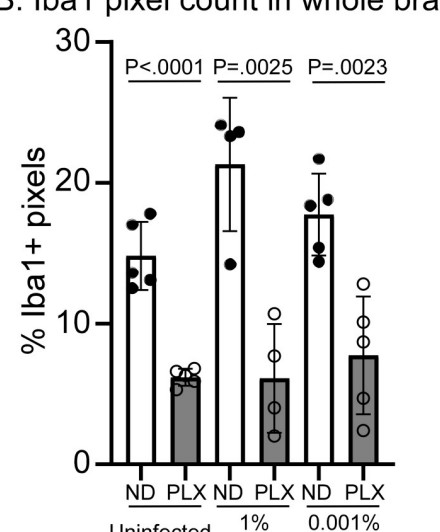

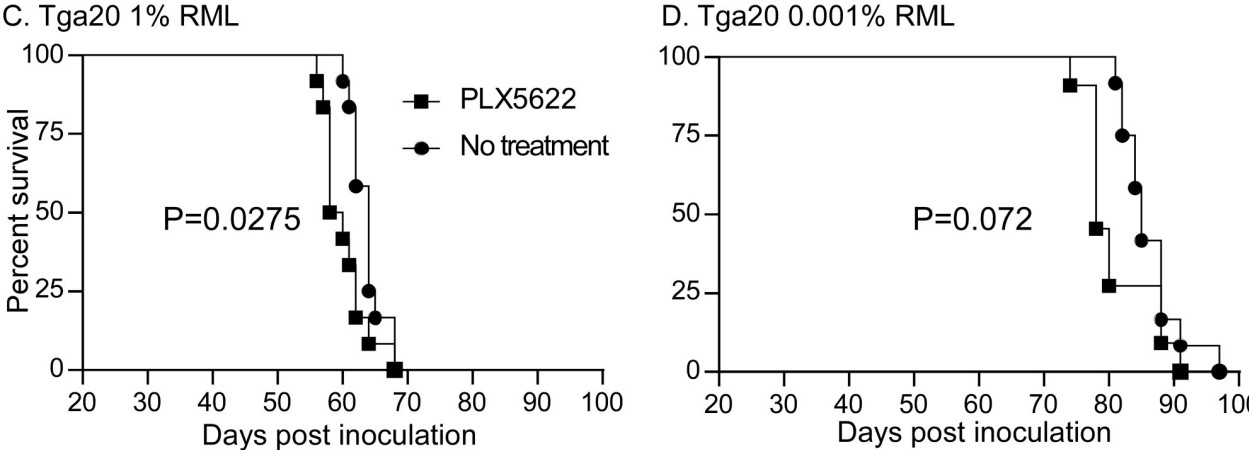

**Fig 5. RML scrapie infection in untreated and PLX5622 treated tga20 mice. A**. Immunoblot comparisons of PrPres accumulation in C57BL/10 mice compared to tga20 mice. Two-fold dilutions of proteinase K treated brain samples were loaded from scrapie infected C57BL/10 or tga20 mice with terminal disease. Estimated molecular weights (in kilodaltons) are shown on the left. Approximate brain equivalents (br. eq.) in milligrams are shown across the bottom. Band intensities were quantified and based on intensity and br. eq. loaded, we calculated that at the point of terminal disease the tga20 mouse brains accumulated approximately 8% of the amount of PrPres as the C57BL/10 mice. **B**. Iba1 positive pixel count comparisons of whole brain sections between untreated (ND) and PLX5622 treated (PLX) tga20 mice from both RML-scrapie infection groups. Untreated and PLX5622 treated mice were compared using unpaired T-tests for each experiment. In all cases, PLX5622 treated mice had significantly fewer microglia. **C, D**. Survival curves following infection with either 1% or 0.001% RML scrapie brain homogenate. Each survival group included 12 mice per group. Statistical analysis (Mantel-Cox, Log-rank test) results are shown within the panels.

## Discussion

Our results support earlier findings that the benefit provided by MG during prion infection likely occurs at the late stages of disease, but the mechanism is unclear. In other chronic neuro-degenerative diseases, activated MG assume a specific transcriptional phenotype termed MGnD or DAM [19, 20]. The elimination of MGnD/DAM can be neuroprotective, leading to the resolution of neuroinflammation and reversal of behavioral impairment [28, 29, 36, 37]. Unlike other neurodegenerative diseases, the ablation of MG during prion-infection is detrimental to the host [22]. Furthermore, MG in the prion-infected brain do not appear to assume

the MGnD/DAM phenotype, rather they assume a distinct transcriptional signature that shares some of the upregulated genes associated with the MGnD subpopulation [21]. The expression of MG-associated immune effectors or interactions with other cells of the CNS might be essential in extending survival in our model. Thus, the activated MG in the prion-infected brain might be attempting to coordinate a neuroprotective response, albeit unsuccessfully due to the overwhelming accumulation of infectious prions associated with astrocytes and neurons [39].

Our findings indicate the presence or absence of MG had little effect on prion clearance at the inoculation site or on reducing prion propagation up to 112 days after infection in C57BL/ 10 mice, which is approximately 70% of the incubation period. In our experiments, macrophages are the dominant cell type adjacent to the needle track by 3 dpi, and one could assume that these professional phagocytes are present to clear debris, including prions. Additionally, astrocytes have been shown to be phagocytic and can compensate when MG phagocytic capabilities are impaired [40–42]. After CNS injury the formation of a glial scar composed primarily of activated astrocytes occurs within hours and decreases after 14 days [43]. Furthermore, astrocytes can take up and propagate prions in vitro [44, 45], and we have reported that astrocytes are more highly activated and PrPres accumulates faster in prion-infected mice with reduced MG [21, 22]. Thus, it is possible that astrocytes likely contribute to early phagocytosis of prions in vivo, which could then lead to astrocytic infection, prion propagation, and spread of infectious prions in the CNS.

Though we conclude that the beneficial functions of MG occur in the later third of the incubation period when PrPres is rapidly accumulating within infected mice, it is possible that MG become immunologically exhausted or less responsive as the disease progresses. Using short-term treatments with PLX5622, the majority of the MG in the CNS can be eliminated and with the removal of the inhibitor MG can quickly repopulate. This repopulation approach has been shown to promote CNS recovery, reduce neuroinflammation, reverse neuronal deficits, and stimulate neurogenesis in mice [35–38, 46]. We treated mice with up to 3 pulses with PLX5622 for a week, and allowed for MG recovery, but found no difference in survival times. These observations were likely confounded by the lack of MG reduction after the week of PLX5622 treatment. In our previous studies with continual long-term PLX5622 treatment, the MG population is substantially reduced but begins to marginally recover during the later phases of prion infection [22]. The reasons for the MG recovery are unknown, but *Csf1R* [21] and *Csf1* [4] are increased in the CNS during prion infection and might alter the pharmacological dynamics of PLX5622 treatment. Perhaps, MG depopulation midway through the prion disease incubation period will require longer or higher-dose treatments. Nevertheless, our intermittent treatment and recovery design was unable to significantly alter the MG population, the disease course, or reactive gliosis.

Lastly, MG reduction in tga20 mice, which express up to 10-fold more PrP$^C$ [31, 32], did not greatly accelerate disease as seen in past studies using C57BL/10 mice. Possibly, the speed at which tga20 mice succumb to prion infection overshadows the potential benefit that MG exhibit. Therefore, only a small yet statistically significant reduction in survival was observed at the highest infectious dose due to the threshold afforded by such a rapid disease onset. Similar results were seen in RML-infected tga20$^{+/+;TK+}$ mice that were treated with ganciclovir to reduce microglia [47]. Previously we proposed that chronically activated MG might be beneficial to the host by removing accumulated PrPres at the later stages of disease [22]. If this is true, then the lower levels of measurable PrPres seen in infected tga20 mouse brains might be insufficient to study the benefit of MG during prion infection.

MG are considered a first line of defense in the CNS since they express numerous damage- and pathogen-associated molecular pattern receptors, phagocytize debris and pathogens, and

secrete proinflammatory immune effectors upon activation [48–50]. Perhaps, microglia are unable to initially identify prions as a pathogenic threat since the infectious conformer is a host-derived protein. This would explain the lackluster inflammatory response when mono-cultures of MG are directly exposed to 1.0% prion-infected brain homogenate in vitro [51], which differ significantly from the cytokine responses attributed to MG in the brains of prion-infected mice [21, 22, 51, 52]. Ultimately, identifying strategies to invoke MG during the early phases of disease might increase their neuroprotective properties, but this will only be achieved if we better understand the molecular mechanisms MG employ to prolong survival during the later stages of prion infection.

## Materials and methods

### Mice

All mice were housed at the Rocky Mountain Laboratory (RML) in an AAALAC accredited facility in compliance with guidelines provided by the Guide for the Care and Use of Laboratory Animals (Institute for Laboratory Animal Research Council). Experimentation followed RML Animal Care and Use Committee approved protocol #2019–013. A total of 201 mice were used to complete all the experiments described. C57BL/10 (C57) mice were used for all experiments except for the tga20 experiment. C57 mice were originally obtained from Jackson Laboratories and have been inbred at RML for many years. Tga20 mice were originally obtained from the European Mouse Mutant Archive and have been partially backcrossed to a C57BL/10 background at RML. Tga20 homozygous mice were used for experiments.

### PLX5622 treatment

Mice were fed purified rodent diet AIN-76A (D10001, Research Diets, Inc.) with or without supplementation with compound PLX5622 (1200 mg/kg chow), kindly provided by Plexxikon Inc., Berkeley, CA. This concentration of PLX5622 in rodent chow has been shown to reduce microglia in the brain by approximately 90% within 10 to 20 days of administration [23]. For the limiting dilution end-point titration experiment, PLX5622 feeding was initiated at day -25 (relative to inoculation at day 0) and stopped at day +60. For the stereotactic experiments, PLX5622 was initiated at day -7 and stopped at day +3 or +7, for the mid-points at days 77 and 112 dpi PLX5622 was initiated at day -7 and continued until mice were euthanized, for the pulse treatment experiments, PLX5622 was fed in three variations of one-week intervals, as shown in Fig 4, for both tga20 experiments, PLX5622 was initiated at day +14 and continued until mice were euthanized for terminal scrapie.

### Intracerebral (ic) inoculations

For all scrapie infections, with the exception of the stereotactically infected mice, mice were anesthetized with isoflurane and then injected in the left-brain hemisphere with 30 microliters of NBH or diluted scrapie brain homogenates. Brain homogenates were diluted in phosphate buffered balanced saline solution + 2% fetal bovine serum. For the limiting dilution end-point titration study, serial 10-fold dilutions of 22L brain homogenate stock from $10^{-6}$ through $10^{-9}$ were inoculated into groups of 5–8 mice per dilution. For the mid-point RT-QuIC timepoints we intracerebrally inoculated groups of 3–4 mice per time-point with 1% 22L containing $6.0 \times 10^5$ LD50 per injection. Groups of 12 tga20 mice were inoculated with either 1% RML containing $2.4 \times 10^4$ LD50 per injection or 0.001% RML containing $2.4 \times 10^1$ LD50 per injection. In each study, groups of PLX5622 treated mice were compared to untreated mice (see PLX5622 section for treatment schedules).

Following inoculation, mice were monitored for onset of prion disease signs by observers blinded to the experimental groups. Once mice showed early signs of clinical disease, monitoring was performed 5–7 times per week. Mice designated for survival studies were euthanized prior to death, when they displayed advanced stages of prion disease including poor grooming, ataxia, subjective weight loss (body condition score < 2), delayed response to stimuli, and prolonged somnolence. For the C57BL/10 mice, typical times from early (subtle) clinical signs to euthanasia range from 4–6 weeks. Tga20 mice have a much shorter clinical duration, typically 3–5 days. In addition, several groups of mice were euthanized at selected time points prior to advanced clinical signs (days 3,7,77,87,108, 112 and 129 dpi) for experiments that did not necessitate a clinical end-point. For the end-point titration study, mice that did not show clinical signs of prion disease were euthanized at 400 dpi. For all experiments, at the time of euthanasia, half of the brain was placed into formalin for future histology and half of the brain was flash frozen for biochemical analysis.

## Stereotactic surgery and microinjection

Mice were anesthetized with isoflurane and positioned on a stereotaxic frame (David-Kopf Instruments, Tujunga, CA). A 1-cm midline incision was made in the skin over the dorsal surface of the skull, and the skull was exposed to allow the positioning of a drill over the bregma point of reference. Two injections were performed for each mouse, one on each side of the brain, directly across from each other and equidistant from midline. To target the striatum, we used the following coordinates from bregma: +1 mm anteroposterior, 1.7 mm lateral, and −2.5 mm ventral to the skull surface. Ten percent brain homogenates (22L scrapie containing $1x10^5$ LD50 / 0.5 μl or normal brain) were injected with Nanofil syringes (World Precision Instruments, Sarasota, FL) and steel bevel needles (33-gauge diameter; World Precision Instruments) into the striatum at a rate of 0.25 μl/min with a total of 0.5 μl per mouse controlled with a pump (UltraMicroPump III with a Micro4 pump controller; World Precision Instruments). The needle was kept in place for 2 min following injection to avoid any reflux of the BH solution. The skin incision was closed with suture. These conditions produced minimal mechanical trauma to the brain. The patency of the needles was verified prior to and after injections. Following surgery, mice received a single 0.2 mg/kg injection of buprenorphine subcutaneously. At either 3 or 7 days post injection, 4 mice per experimental group were euthanized by isoflurane anesthesia overdose, followed by cervical dislocation. After euthanasia, one half of the brain was removed and placed in formalin. The second half of the brain was used to collect tissue surrounding the needle track for RT-QuIC assays. A 3.5 mm diameter biopsy instrument was used to remove a cylindrical region around the needle track for these samples.

## RT-QuIC assay and SD50 calculations

We quantified PrP amyloid seeding activity present in brain homogenates from 22L infected mice at 3, 7, 77 and 112 dpi by RT-QuIC assay. Ten percent brain homogenates were centrifuged at $2,000 \times g$ for 2 min to remove large particulates. The remaining supernatants were aliquoted and frozen at −80˚C for RT-QuIC analysis at a later time. After thawing, supernatants were serially diluted in half-log ($10^{-0.5}$) increments with PBS, 0.05% SDS, and N2 medium supplement (1×; Gibco). Dilutions tested ranged from $10^{-2.5}$ to $10^{-6.5}$ depending on the expected endpoint. Reactions were performed as previously described using recombinant mouse PrP (0.1 mg/ml final concentration) as a substrate [34]. Four wells were subjected to RT-QuIC for each dilution. Each 100-μl reaction mixture was seeded with 2 μl of diluted sample, resulting in a final SDS concentration of 0.001% during incubation. The number of positive wells/total number of wells tested at each dilution for each sample was determined based on ThT

fluorescence at 50 h, and these values were used to calculate the seeding activity. Positive wells were defined based on an increased fluorescence value >21 SD above the mean of all negative controls. $SD_{50}$ calculations were performed for each sample using the Spearman-Karber formula for titer calculation.

## Immunoblotting and PrPres comparisons

Immunoblotting for PrPres was performed to confirm disease in the end-point titration mice and to compare PrPres levels between RML scrapie infected C57BL/10 and tga20 mice. Brain homogenization, proteinase K (50 mg/ml) treatment, gel electrophoresis, protein transfer and immunoblotting with anti-PrP antibody D13 were performed as previously described [22] with the following modifications. In the current immunoblots the secondary antibody was peroxidase-conjugated sheep anti-human IgG (Sigma) at a 1:10,000 dilution. Gel wells for disease confirmation were loaded with 0.6 mg brain equivalents (br.eq) per lane, C57BL/10 and tga20 PrPres level comparison gels were loaded with several different dilutions of protein ranging from 0.06–0.96 mg br.eq to allow for comparison.

Protein bands were visualized using an enhanced chemiluminescence (ECL) detection system (GE Healthcare) and exposure to either film or imaged using an iBright FL1000 Imaging System (Invitrogen) capable of capturing blot images from a membrane exposed to ECL and utilizing the "Chemiluminescent blot" imaging protocol. For comparison of PrPres levels between RML-scrapie infected C57BL/10 and tga20 mice, we compared band intensities between serial two-fold dilutions of brains from either two tga20 or two C57BL/10 mice. Band intensities were obtained by taking images of exposed film with a ChemiDocMP Imaging System (BioRad) with a white-light transilluminator plate, and analysis of band intensities was performed using the ImageLab 6.1 software (Bio-Rad.) Using ImageLab Analysis Tools, lanes were manually selected, and bands were detected automatically using low sensitivity. Background subtraction was performed using the analysis function and was consistent across all lanes for each gel, and the Lane Profile tool was used to ensure the entirety of each band was represented. Once captured, the numerical data was transferred to Excel and using the known brain equivalents loaded per lane and the corresponding band intensity data, we calculated the different levels of PrPres accumulation between C57BL/10 and tga20 mice.

## Immunohistochemical and H&E staining

To analyze histopathology, we performed serial sections through the needle track and performed IHC for either microglia and macrophages (Iba1), microglia (TMEM119), astrocytes (anti-GFAP) or prion protein (D13). Slides were stained with a standard protocol of hematoxylin and eosin (H&E) for observation of the overall pathology.

For each experimental group 3–5 half-brains were removed and placed in 10% neutral buffered formalin for 3 to 5 days. Tissues were then processed by dehydration and embedding in paraffin. Sections were cut using a standard Leica microtome, placed on positively charged glass slides, and air-dried overnight at room temperature. The following day slides were heated in an oven at 60˚C for 20–30 min.

For all IHC, de-paraffinization, antigen retrieval and staining were performed using the Discovery XT Staining Module. Antigen retrieval for D13 staining was achieved using extended cell conditioning with CC1 buffer (Ventana) containing Tris-Borate-EDTA, pH 8.0 for 100 minutes at 95˚C. To stain PrP we applied D13, a monoclonal anti-PrP antibody [22], at a dilution of 1:100 in antibody dilution buffer (Ventana) for 2 hours at 37˚C. The secondary antibody, biotinylated goat anti-human IgG (Jackson ImmunoResearch, West Grove, PA) was

diluted 1:250 in Ventana antibody dilution buffer and applied for 32 min at 37˚C. Detection was performed with ChromoMap DAB (Roche/Ventana #NC1859896).

Antigen retrieval for TMEM119 staining was achieved using extended cell conditioning with CC1 buffer (Ventana) described above for 64 minutes at 95˚C. To stain microglia, we applied polyclonal rabbit antibody TMEM119 (Synaptic systems #400 002) at a dilution of 1:1,000 in antibody dilution buffer (Ventana) for 20 minutes at 37˚C. The secondary antibody, rabbit link (Biogenex #HK336-9R), was applied neat for 1 hour at 37˚C. Detection was performed with ChromoMap DAB (Roche/Ventana #NC1859896).

For detection of microglia the Discovery XT staining system (Ventana Medical Systems) was also used, but antigen retrieval was milder (see below) and staining used a RedMap detection kit and hematoxylin counterstain. To stain microglia and macrophages, anti-Iba1 antiserum was generated by immunization of rabbits with a 14 amino acid peptide from the C-terminus of the Iba1 protein as previously described [53] and was a generous gift from Dr. John Portis. For Iba1, antigen retrieval was done using the standard CC1 protocol (cell conditioning buffer containing Tris-Borate-EDTA, pH 8.0, ~ 44 min at 100˚C). Anti-Iba1 was used at a 1:2000 dilution and applied for 40 min at 37˚C. The secondary antibody was biotinylated goat anti-rabbit IgG (Biogenex Ready-to-use Super Sensitive Rabbit Link) and was applied for 40 min @ 37˚C. The secondary antibody was biotinylated goat anti-rabbit IgG described above and was applied for 16 min at 37˚C. Images were scanned and photographed using Aperio eSlide Manager and Imagescope software (Leica).

## Pixel counting and cell counting

Iba1 stained slides were examined and photographed using Image Scope software. Iba1 cells were quantified by two different methods, either manual counting from photographs of cerebral cortex and thalamus, or from pixel counting brain sections using Image Scope. Manually counted cells were reported as Iba1 positive cells per $mm^2$ of brain. Iba1 was also quantified using the ImageScope positive pixel count algorithm (version 9.1). For each stained brain section, a 5-micron thick median sagittal section representing approximately 55 $mm^2$ was evaluated at 1× magnification. Data was reported as the percentage of positive pixels (positive pixels/total pixels × 100) in the section. Comparisons of the manually counted numbers of Iba1-positive microglia and pixel count data between PLX5622-treated and untreated cohorts were made by unpaired Student's $t$ test using GraphPad Prism.

## Survival curve statistical analysis

Statistical curve analysis (Mantel-Cox test) for the intermittent PLX5622 treatment study and the tga20 mouse experiments were performed using GraphPad Prism software.

## Supporting information

**S1 File.**
(XLSX)

**S1 Raw image.**
(TIF)

## Acknowledgments

We thank Drs. Cathryn Haigh, Suzette Priola, and Clayton Winkler for critical review of the manuscript, and Jeff Severson for animal husbandry. Compound PLX5622 was kindly provided by Plexxikon Inc.

## Author Contributions

**Conceptualization:** Brent Race, James F. Striebel, Bruce Chesebro, James A. Carroll.

**Data curation:** Brent Race, Katie Williams, Chase Baune, Dan Long, Tina Thomas, Lori Lubke, James A. Carroll.

**Formal analysis:** Brent Race, James F. Striebel, James A. Carroll.

**Investigation:** Brent Race.

**Methodology:** Bruce Chesebro.

**Resources:** Bruce Chesebro.

**Supervision:** Brent Race, Bruce Chesebro.

**Writing – original draft:** Brent Race, James A. Carroll.

**Writing – review & editing:** Brent Race, Katie Williams, Chase Baune, James F. Striebel, Bruce Chesebro.

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
