## [Decision Letter · Decision Letter 0]

9 Aug 2022

PONE-D-22-15186Microglia have limited influence on early prion pathogenesis, clearance, or replicationPLOS ONE

Dear Dr. Race,

Thank you for submitting your manuscript to PLOS ONE. After careful consideration, we feel that it has merit but does not fully meet PLOS ONE’s publication criteria as it currently stands. Therefore, we invite you to submit a revised version of the manuscript that addresses the points raised during the review process.

We look forward to receiving your revised manuscript.

Kind regards,

Human Rezaei

Academic Editor

PLOS ONE

Journal Requirements:

“This research was supported by the Intramural Research Program of the NIH, National Institute of Allergy and Infectious Diseases.”

Reviewers' comments:

Reviewer's Responses to Questions

**Comments to the Author**

1. Is the manuscript technically sound, and do the data support the conclusions?

Reviewer #1: Yes

Reviewer #2: Partly

2. Has the statistical analysis been performed appropriately and rigorously? 

Reviewer #1: Yes

Reviewer #2: No

3. Have the authors made all data underlying the findings in their manuscript fully available?

Reviewer #1: Yes

Reviewer #2: Yes

4. Is the manuscript presented in an intelligible fashion and written in standard English?

Reviewer #1: Yes

Reviewer #2: Yes

5. Review Comments to the Author

Reviewer #1: Race et al report on the effects of pharmacologically-induced microglia depletion on in vivo prion pathogenesis.

This is a follow-up of initial report showing that long-term PLX5622-induced microglia depletion during the pathogenesis significantly accelerated prion disease tempo (10.1128/JVI.00549-18). The objective here was to narrow the phase during which microglial depletion had an impact on prion replication.

Four different experimental paradigms were used. They lend support for the i), absence of significant impact of microglia depletion on the early stage of prion infection (i.e., absence of impact on prion clearance or erly replication, ii) absence of impact at mid-stage when performed during a short period of time.

Overall, the authors suggest that microglia impact prion disease in vivo mostly during the last third of the disease.

I have no specific comments. The study is well-done, with ad hoc controls.

Reviewer #2: Comments on the MS entitled « Microglia have limited influence on early prion pathogenesis, clearance, or replication” and submitted for publication on PlosOne (ref #PONE-D-22-15186), written by Brent Race, D.V.M, Katie Williams, Chase Baune, James F. Striebel, Dan Long, Tina Thomas, Lori Lubke, Bruce Chesebro and James A. Carroll.

The paper describes prion replication in the brains of mice whose microglia (MG) has been depopulated (pharmacological treatment using PLX5622 drug) at different stages before or after inoculation with 22L prion strains. The rationale for this approach is that early drug treatment and MG depletion after Prion infection significantly accelerates clinical disease. This results have been obtained earlier in the laboratory and suggest a protective role of MG in the disease course (up to 30 days of life expectancy in this particular study). This result prompted the authors to investigate in further details MG contribution: first their contribution in early disease phase; in a second series of experiments, the authors investigated the effects of repeated MG depopulation during the course of a prion infection. Ultimately, since high PrPC expression levels resulted in accelerated disease but associated less PrPSc depots, the role of basal level of prion expression has been investigated in the context of MG depopulation in either high PrP expressor (tga20 mice) versus wild-type mice.

The main findings of this paper is that MG depletion is of little impact on prion replication, whatever the depopulation occurs in early phase or repeated conditions, or even when high PrPC expression levels are considered.

The experiments are precisely described, and the authors conclude in the absence of MG influence on the course of the prion disease. However, these main results have been obtained in a context where:

- the incubation time is only marginally reduced in treated vs control animals (as compared with the nearly 3 weeks reduction observed in the work by Carroll et al., ref 22);

- only 10-25% of the MG are indeed depopulated as compared with the 90% reduction of the previous study (at least in the last two experiments, cf MS lines 205-6, figure 4 C-E)

- the seeding activity, as evaluated using RT-QuIC is statistically the same in both groups.

This set of data suggests that the MG depletion did not occur as it has in the previous report. The only point that is, in my opinion, suggestive of a MG reduction in the treated animals during the first experiment is the staining of TMEM119 (shown as a photo in fig 1), since this labeling is more specific to MG (cf figure 1 panels J and K). However the authors do not present any quantitative data regarding this reduction.

Since MG depletion did not occur as expected, I wonder why the authors did not comment further on the administration process, that may have not been appropriate for the required depletion timing. With respect to the drug administration modes, may the differences between the three experiments account for the differential efficacy of MG depletion? These differences are sufficiently significant to be discussed.

I do not completely agree with the initial statement of the discussion in which the authors pledge that “the benefit provided by MG during prion infection likely occurs at the late stages of disease”. This is not demonstrated in this MS, but rather in the previous ref 22. In the same way, since MG repopulation is proposed to occur after 60 dpi (end of the early PLX5622 treatment), and therefore is suspected to be responsible for the overall lack of efficiency of the treatment, a control experiment in which treatment would have been administered all along should have been set up.

Therefore I suggest the authors either amend their data or their discussion.

Minor comment :

The meaning of sentence L173-176 is difficult to catch

In the RT-QuIC M&M section, since it is mentioned for every other component, please precise PrP concentration.

6. PLOS authors have the option to publish the peer review history of their article (what does this mean?). If published, this will include your full peer review and any attached files.

Reviewer #1: No

Reviewer #2: **Yes: **pierre sibille

---

## [Author Response · Author response to Decision Letter 0]

1 Sep 2022

Please see attached cover letter and response to reviewers for detailed descriptions

---

## [Decision Letter · Decision Letter 1]

17 Oct 2022

Microglia have limited influence on early prion pathogenesis, clearance, or replication

PONE-D-22-15186R1

Dear Dr. Race,

We’re pleased to inform you that your manuscript has been judged scientifically suitable for publication and will be formally accepted for publication once it meets all outstanding technical requirements.

Kind regards,

Human Rezaei

Academic Editor

PLOS ONE

Additional Editor Comments (optional):

Reviewers' comments:

Reviewer's Responses to Questions

**Comments to the Author**

1. If the authors have adequately addressed your comments raised in a previous round of review and you feel that this manuscript is now acceptable for publication, you may indicate that here to bypass the “Comments to the Author” section, enter your conflict of interest statement in the “Confidential to Editor” section, and submit your "Accept" recommendation.

Reviewer #1: All comments have been addressed

Reviewer #2: All comments have been addressed

2. Is the manuscript technically sound, and do the data support the conclusions?

Reviewer #1: Yes

Reviewer #2: Yes

3. Has the statistical analysis been performed appropriately and rigorously? 

Reviewer #1: Yes

Reviewer #2: No

4. Have the authors made all data underlying the findings in their manuscript fully available?

Reviewer #1: Yes

Reviewer #2: Yes

5. Is the manuscript presented in an intelligible fashion and written in standard English?

Reviewer #1: Yes

Reviewer #2: Yes

6. Review Comments to the Author

Reviewer #1: In this revised version, the authors have addressed the referees concerns satisfactorily and the manuscript is now acceptable for publication.

Reviewer #2: Parametric statistical tests are not adapted for datasets of fewer than 20 replicates. Non-parametric tests such as mann-Whitney or Kruskall-Wallis are preferred

7. PLOS authors have the option to publish the peer review history of their article (what does this mean?). If published, this will include your full peer review and any attached files.

Reviewer #1: No

Reviewer #2: No

---

## [Editor Report · Acceptance letter]

19 Oct 2022

PONE-D-22-15186R1 

Microglia have limited influence on early prion pathogenesis, clearance, or replication 

Dear Dr. Race:

I'm pleased to inform you that your manuscript has been deemed suitable for publication in PLOS ONE. Congratulations! Your manuscript is now with our production department. 

Kind regards, 

on behalf of

Dr. Human Rezaei 

Academic Editor

PLOS ONE